# The Current View of Nonalcoholic Fatty Liver Disease-Related Hepatocellular Carcinoma

**DOI:** 10.3390/cancers13030516

**Published:** 2021-01-29

**Authors:** Tomomi Kogiso, Katsutoshi Tokushige

**Affiliations:** Department of Internal Medicine, Institute of Gastroenterology, Tokyo Women’s Medical University, 8-1 Kawada-cho, Shinjuku-ku, Tokyo 162-8666, Japan; tokusige.katsutoshi@twmu.ac.jp

**Keywords:** nonalcoholic fatty liver disease, hepatocellular carcinoma, prevalence, prognosis, surveillance

## Abstract

**Simple Summary:**

The incidence of nonalcoholic fatty liver disease (NAFLD)-related hepatocellular carcinoma (HCC) is increasing. However, an effective screening or surveillance method is not established. Recently, the NAFLD/nonalcoholic steatohepatitis (NASH) guidelines of Japan were revised to incorporate new strategies and evidence for the management and surveillance of NAFLD/NASH. Advanced fibrosis and lifestyle-related and metabolic comorbidities, especially obesity and diabetes mellitus, are associated with HCC development. At the first screening, serum markers of hepatic fibrosis (hyaluronic acid, type IV collagen 7S, and mac-2 binding protein), or the fibrosis (FIB)-4 index or the nonalcoholic fatty liver disease fibrosis score (NFS), or a platelet count should be evaluated. When liver fibrosis is indicated, consultation with a gastroenterology specialist should be considered for the second screening. The risk of HCC should be stratified using the FIB-4 index or the NFS. Liver stiffness should be measured using vibration-controlled transient elastography in those at intermediate or high risk. Blood tests and imaging should be performed every 6–12 months in patients with advanced fibrosis for HCC surveillance. We review here what is known about NAFLD-HCC and provide perspectives for future research.

**Abstract:**

Nonalcoholic fatty liver disease (NAFLD) is the hepatic manifestation of metabolic syndrome and can develop into hepatocellular carcinoma (HCC). The incidence of NAFLD-related HCC, which is accompanied by life-threatening complications, is increasing. Advanced fibrosis and lifestyle-related and metabolic comorbidities, especially obesity and diabetes mellitus, are associated with HCC development. However, HCC is also observed in the non-cirrhotic liver. Often, diagnosis is delayed until the tumor is relatively large and the disease is advanced; an effective screening or surveillance method is urgently required. Recently, the NAFLD/nonalcoholic steatohepatitis (NASH) guidelines of Japan were revised to incorporate new strategies and evidence for the management and surveillance of NAFLD/NASH. Fibrosis must be tested for noninvasively, and the risk of carcinogenesis must be stratified. The treatment of lifestyle-related diseases is expected to reduce the incidence of NAFLD and prevent liver carcinogenesis.

## 1. Introduction

The prevalence of nonalcoholic fatty liver disease (NAFLD) has been growing worldwide, and the incidence of this disease in Japan is estimated to be 29.7% [1]. NAFLD may give rise to hepatocellular carcinoma (HCC); such HCC is becoming more common [2]. However, the characteristics of, risk factors for, and prognosis of NAFLD-HCC have not been elucidated fully. The American Association for the Study of Liver Diseases [3], European Association for the Study of the Liver [4], and American Gastroenterological Association (AGA) [5] have published practice guidelines for NAFLD management. Recently, the Japanese NAFLD/nonalcoholic steatohepatitis (NASH) guidelines [6] were revised to include new evidence and strategies for NAFLD/NASH management and surveillance. Here, we review what is known about NAFLD-HCC and provide perspectives for future research.

## 2. Prevalence of NAFLD-HCC

In a multicenter survey performed in Japan, Tateishi et al. [2] examined 7370 patients with HCC and other liver diseases and reported that the incidence of non-B non-C liver cancer increased from 10% in 1991 to 32.5% in 2015. Such liver cancer was induced by alcoholic liver disease (ALD) in 675 (32.3%) patients, NAFLD/NASH in 315 (15.1%) patients, and unknown causes in 911 (43.7%) patients. Such “unknown cause(s)” may include “burned out” NASH [7]. Thus, the prevalence of NAFLD-HCC may be underestimated.

Reports on HCC development after cirrhosis caused by NAFLD are listed in Table 1. Yatsuji et al. [8] described cancers that developed in 7 of 68 patients with NASH and cirrhosis (average age 63 years, 57% male) over an average follow-up period of 3.4 years. The five-year cumulative carcinogenesis rate was 11.3%, 30.5% that of the control group [patients with hepatitis C virus (HCV)-related cirrhosis]. Ascha et al. [9] studied 195 patients with NASH cirrhosis (mean age 56.6 years, 44.1% male) for a median of 3.2 years, and found liver cancer in 25 (12.8%) of these patients; the annual rate was 2.6%. In another report, the annual HCC rate in patients with liver cirrhosis was 1–3% [10]. The annual HCC rate in patients with liver cirrhosis caused by HCV was 6% [11]; the rate of HCC associated with NAFLD was lower. In one large study, the NAFLD-HCC rate was 0.44/1000 person-years [95% confidence interval (CI) 0.29–0.66]; it was 5.29/1000 person-years (95% CI 0.75–37.56) in patients with NASH [12]. As mentioned above, the HCC rate is not high, but the number of patients with cancer is increasing, given the large population with NAFLD. In Japan, about 23 million persons are thought to have NAFLD and 100,000 will develop HCC over the next 10 years. This situation cannot be ignored.

Cancer in non-cirrhotic subjects has also been reported (Table 2). Kawada et al. [13] evaluated 1168 surgically resected HCCs; 6 were not associated with cirrhosis. In a study of 1562 patients, Bengtsson et al. [14] found that 83 of 225 NAFLD-HCCs were not associated with cirrhosis. HCC arising in non-cirrhotic livers constituted 10–50% of all cases and was thus more common than HCV-related HCC [15]. A multicenter study of 209 patients with liver cirrhosis reported that F4 HCC developed in 72.7% of females and 37.6% of males; a sex difference was thus evident [16]. In contrast, the frequency of HCC in non-cirrhosis was reported to be significantly higher in female than male [17]. Thus, HCC also should be noticed in non-cirrhotic cases.

## 3. Characteristics and Pathogenesis of NAFLD-HCC

The “multiple parallel hits” hypothesis has been advanced to explain NASH onset [42]. Adipose tissue lipotoxicity, insulin resistance, lifestyle-related diseases, alcohol consumption, growth and sex hormone–related factors, menopause, aging, oxidative stress (disruption of iron and free fatty acid levels), and changes in the profiles of gut and oral bacteria and Helicobacter pylori are thought to contribute to NASH pathology and carcinogenesis. The hyperinsulinemia associated with insulin resistance may promote cell proliferation and trigger carcinogenesis. Cellular stress responses, including autophagy and endoplasmic reticulum-associated activity, may trigger NAFLD cytotoxicity [43]. In addition, innate immune reactions between the products of intestinal bacteria and the liver increase deoxycholic acid levels, triggering the senescence of hepatic stellate cells [44]. The senescence-associated secretory phenotype, which features the extracellular secretion of inflammatory cytokines (tumor necrosis factor-α and interleukin-6), chemokines, and extracellular matrix–degrading enzymes from senescent cells, may trigger liver carcinogenesis [45]. Furthermore, changes in the profiles of hepatokines, including angiopoietin-like factors and fibroblast growth factors [46], and acylcarnitine may be involved in carcinogenesis [47]

Genetic influences on NASH and carcinogenesis have been reported. The gene encoding the patatin-like phospholipase domain-containing 3 (PNPLA3) protein is widely accepted to affect disease susceptibility. The frequency of “risky” alleles is elevated in patients with liver cancer; the gene is thus presumed to be associated with carcinogenesis [48]. Additionally, 17-beta hydroxysteroid dehydrogenase 13 (HSD17B13) was identified as having the protective role of NAFLD [49] and protecting against HCC development [50]. Lower HSD17B13 in non-cancerous tissues was associated with worse recurrence-free survival and overall survival in HCC patients. Although they mainly studied viral hepatitis-associated HCC, further study is required.

Like similar liver diseases, NAFLD-HCC is characterized by a trabecular phenotype [15]. In patients with a common subtype of NAFLD-HCC, Salomao et al. [51] reported cancers with the morphological characteristics of steatohepatitis and termed this phenotype steatohepatitic HCC (SH-HCC). The expression of markers of hepatocellular adenoma is not specific to NAFLD-HCC; but is evident in some patients with NAFLD-HCC [52]. Notably, the levels of serum amyloid A and c-reactive protein were increased significantly in patients with SH-HCC. In addition, HCC tumors in non-cirrhotic livers are large, and may have arisen from hepatocellular adenomas [31].

Turning to tumor markers, in a study of 209 cases in Japan, the α-fetoprotein (AFP) positivity rate was about 50% (AFP 10–100 ng/mL, 27.9%; ≥100 ng/mL, 20.6%), accompanied by an abnormal prothrombin [des-gammacarboxy prothrombin (DCP)]. The DCP positivity rate was about 68% (DCP 40–100 mAU/mL, 11.8%; ≥100 mAU/mL, 56.4%). This rate was slightly higher in patients with NAFLD-HCC than in others [16].

In terms of HCC in the non-cirrhotic liver, Mohamad et al. [37] found that patients with HCC who lacked cirrhosis were older than those with cirrhosis [67.5 ± 12.3 vs. 62.7 ± 8.1 years], less likely to be obese (52% vs. 83%) or to have type 2 diabetes (38% vs. 83%), more likely to have single nodules (80.6% vs. 52.2%) or larger (>5 cm) nodules (77.8% vs. 10.6%) and to undergo hepatic resection (66.7% vs. 17%), and less likely to receive locoregional therapy (22.3% vs. 61.7%) or deceased-donor liver transplantation (LT; 0% vs. 72.3%). Kodama et al. [39] described HCCs in non-cirrhotic livers as large; recurrence after curative surgical treatment was less common than in patients with HCC and cirrhosis.

## 4. Risk Factors for NAFLD-HCC

In a study of 622 patients with decompensated cirrhosis, Muto et al. [53] found that the risk factors for liver cancer were male sex, diabetes, high body mass index (BMI), AFP level ≥ 20 ng/mL, and low serum albumin level. A 16-year prospective follow-up study of 900,000 persons in the United States showed that among patients with BMIs ≥35 kg/m^2^, the relative risk of death from HCC was 4.52 (95% CI 1.13–2.05) in males and 1.62 (95% CI 1.40–1.87) in females [54]. Obesity is a risk factor for HCC and HCC-related death. Kawamura et al. [55] evaluated 6508 patients aged ≥ 60 years with NAFLD, aspartate aminotransferase (AST) levels ≥40 IU/L, and platelet counts <15 × 10^4^/μL, and found that diabetic complications were independent risk factors for HCC [hazard ratio (HR) 3.21, 95% CI 1.09–9.50, *p* = 0.035] (Table 3). In a study of combined lifestyle-related diseases, the presence of both hypertension and dyslipidemia was associated with a 1.8-fold increase in HCC risk (HR 1.8, 95% CI 1.59–2.06); in the presence of diabetes, obesity, and dyslipidemia, the increase was 2.6-fold (HR 2.6, 95% CI 2.3–2.9) [41]. Kogiso et al. [56] demonstrated that risk factors for new—onset HCC were AST level (HR 1.021, 95% CI 1.009–1.034, *p* = 0.001), platelet count (HR 0.881, 95% CI 0.801–0.970, *p* = 0.010), and treatment for hypertension (HR 4.986, 95% CI 1.223–20.329, *p* = 0.025). Patients with platelet counts <11.5 × 10^4^/μL exhibited significantly greater mortality and a greater risk of HCC development (*p* < 0.01), and the above-listed factors were predictive of HCC development in patients with NAFLD.

In a systematic review, NAFLD and NASH cohorts lacking cirrhosis were at minimal risk of HCC development (cumulative HCC mortality 0–3% over study periods of up to 20 years) [59]. Cohorts with NASH and cirrhosis were at consistently higher risk (cumulative incidence 12.8% over three years and 2.4% over seven years). Tokushige et al. [57] reported that risk factors for NAFLD-HCC in patients with cirrhosis were Child-Pugh score (HR 3.09, 95% CI 1.374–6.934), serum γ-glutamyltranspeptidase level (HR 1.01, 95% CI 1.002–1.022), and older age (HR 1.12, 95% CI: 1.014–1.226). Thus, the evaluation of cirrhosis is important when considering the risk of HCC development in patients with NAFLD.

As risk factors for HCC development in non-cirrhotic livers, Tobari et al. [40] identified male sex [odds ratio (OR) 7.774, 95% CI 2.176–27.775], light drinking (OR 4.893, 95% CI 1.923–12.449), and a high Fibrosis-4 (FIB-4) index (OR 2.634, 95% CI 1.787–3.884). The recurrence rate was significantly lower in patients with NAFLD-HCC who lacked cirrhosis (*p* < 0.01) [39]. The risk factors for recurrence were male sex, lower serum albumin levels, and advanced fibrosis. In females, NAFLD-HCC develops only in cirrhotic livers; it also develops in non-cirrhotic livers in males.

## 5. Treatment of NAFLD-HCC

NAFLD-HCC treatment is determined by liver function and the extent of HCC progression, in accordance with the Liver Cancer Practice Guidelines [60,61] and Barcelona Clinic Liver Cancer staging [62]. However, postoperative mortality is higher in patients with NAFLD-HCC than in those with HCV-HCC [62]. Vascular lesions (complications of lifestyle-related diseases) may be involved.

In patients with end-stage liver disease, NAFLD-HCC is an indicator for LT; NAFLD/NASH has been the fastest growing indicator for LT over the past 20 years [63,64]. Wong et al. [24] studied 61,868 patients who had undergone LT, including 10,061 HCC cases; the incidence of NASH-HCC increased from 8.3% in 2002 to 13.5% in 2012. About 50% of transplant recipients with NASH develop recurrent NAFLD, but the post-LT outcomes of patients with NAFLD are similar to those of patients without NAFLD [65].

## 6. Prognosis of NAFLD-HCC

In a large NAFLD cohort study, the liver-related death rate was 0.77/1000 person-years (95% CI 0.33–1.77) and the overall mortality rate was 15.44/1000 person-years (95% CI 11.72–20.34) [12]. In a NASH cohort study, the liver-related death rate was 11.77/1000 person-years (95% CI 7.10–19.53) and the overall mortality rate was 25.56/1000 person-years (95% CI 6.29–103.8). Eguchi et al. [66] reported a mortality rate of 40.0% over 2.7 years in a Japanese cohort. On long-term follow-up (7.7 years), mortality was higher in F3/F4 than in F0/F2 patients (25.0% vs. 0.0%) [66]. Younossi et al. [27] reported that NAFLD-HCC had a poor prognosis because HCC was diagnosed at an advanced stage. In contrast, NAFLD-HCC–related mortality did not differ significantly by HCC stage in another study [67]. The prognosis of NASH-HCC is reportedly similar to that of ALD-HCC (5-year survival rates, 49.1% and 43.7%) [16]. Notably, the overall survival rate was significantly higher for patients with NAFLD-HCC than for those with hepatitis B virus–HCC and HCV-HCC (HR 0.35, 95% CI 0.15–0.80 and HR 0.37, 95% CI 0.17–0.77, respectively) [68]. The five-year recurrence rate was 69.6% for NAFLD-HCC; DCP was a risk factor [16]. Summary of NAFLD-HCC outcomes after curative therapies were shown in Table 4. Thus, the prognosis of NAFLD-HCC does not differ significantly from those of other HCCs, but early diagnosis and treatment are essential.

## 7. Prevention of NAFLD-HCC

A meta-analysis of the effects of drugs used to treat lifestyle-related conditions on carcinogenesis showed that metformin taken by patients with type 2 diabetes was associated with a reduced risk of HCC [73]. In a large national cohort study, Kaplan et al. [74] found that metformin use was associated independently with a decline in overall mortality, but not liver-related mortality, HCC, or decompensation, in patients with cirrhosis. A large meta-analysis of 4298 HCC cases showed that statins reduced the HCC incidence by 37%, and that the treatment of lifestyle-related diseases may suppress HCC [75].

Recent analysis demonstrated that bariatric surgery was reduced NASH-HCC [76]. Bariatric group showed lower incidence of new-onset HCC (0.05% vs. 0.34%, *p* = 0.03) comparing to the propensity-matched control group. It might become an option for HCC prevention.

## 8. NAFLD-HCC Screening and Surveillance

NAFLD/NASH-HCC is often diagnosed when the tumor is relatively large and the disease is thus advanced [15]. Bertot et al. [77] reported that patients with incidentally diagnosed NAFLD cirrhosis exhibited decreased platelet counts and international normalized ratios (both *p* < 0.05) and were more likely to have HCC (12%). In contrast, a surveillance group (who underwent liver ultrasound every six months) had no HCC complication. Screening and surveillance are necessary. As the cancer risk associated with NAFLD/NASH is minimal, regular surveillance of low-risk groups is inefficient and uneconomical. The risk factor for liver carcinogenesis in patients with NAFLD/NASH is advanced liver fibrosis; such patients require strict follow-up by a general physician applying the algorithm of the revised Japanese guidelines (Figure 1) [6]. At the first screening, serum markers of hepatic fibrosis (hyaluronic acid, type IV collagen 7S, and mac-2 binding protein), or the FIB-4 index or the nonalcoholic fatty liver disease fibrosis score (NFS), or a platelet count should be evaluated. When liver fibrosis is indicated, consultation with a gastroenterology specialist should be considered for the second screening.

Castera et al. [78] found that such simple, inexpensive, and widely available assays had high negative predictive values, effectively ruling out advanced fibrosis. Patients at low risk of advanced fibrosis (FIB-4 score < 1.3 or the NFS score < −1.455) do not require further assessment. Liver stiffness should be measured using vibration-controlled transient elastography in those at intermediate (FIB-4 score 1.3 to 2.66 or NFS score −1.455 to 0.674) or high (FIB-4 score ≥ 2.67 or NFS score ≥ 0.675) risk. Although no NAFLD/NASH-HCC surveillance method has been established, blood tests and imaging should be performed every 6–12 months in patients with advanced fibrosis.

Recently, the AGA proposed an HCC surveillance method for patients with NAFLD and liver cirrhosis, or severe fibrosis revealed by noninvasive testing [5]. When the quality of ultrasonography is suboptimal for screening of HCC, future screening by either computed tomography (CT) or magnetic resonance imaging (MRI) every six months are recommended. The GALAD score (based on age, sex, and AFP, AFP-L3, and DCP levels) is also used to estimate fibrosis [79]. In terms of imaging, abdominal ultrasonography is cost effective and noninvasive, but its diagnostic sensitivity is poor in obese patients [80]. Although CT is useful, patients with NAFLD often have renal disorders contraindicating the use of contrast media, and antidiabetic drugs (especially biguanide) must be stopped a few days before contrast enhancement. Although MRI detects HCC with high sensitivity, it is costly and its contraindications include pacemaker presence and renal dysfunction. The establishment of an efficient, cost-effective method for the screening of those at high risk of HCC is desirable.

## 9. Perspectives for Future Research

The development of tyrosine kinase inhibitors and immune checkpoint inhibitors affected a paradigm shift in HCC treatment [81,82]. However, the molecular and biological mechanisms underlying NAFLD-HCC development are not fully understood. Shimada et al. [83] classified HCCs by reference to genetic and immune profiles. Proteins that play key roles in HCC development would be valuable biomarkers for NAFLD-HCC screening. Molecular profiling might reveal target molecules for NAFLD-HCC therapy. It might be able to apply for personalized therapy and also when loco-regional treatment is not feasible. In addition, extracellular vesicles (EVs) [84,85,86] and CRISPR-Cas9– mediated genomic editing [87] are new tools for cancer therapy. EVs have shown great potential as drug delivery systems; exosomal micro-RNAs from HCC cells enhance transformed cell-like growth of recipient cells. Genomic editing of genes such as PNPLA3 may potentially treat NAFLD-HCC.

In terms of NAFLD-HCC prevention, metformin and pioglitazone reduced NAFLD-HCC development in patients with type 2 diabetes; statins were effective in those with dyslipidemia. Several novel NAFLD treatment agents are under evaluation. Future studies will explore new monotherapies and combination therapies.

## 10. Conclusions

The incidence of NAFLD-HCC will increase as obesity increases. Efficient surveillance is essential. Fibrosis should be evaluated noninvasively and the risk of cancer should be stratified. The treatment of lifestyle-related diseases prevents liver carcinogenesis. Further studies should seek to identify more noninvasive markers of fibrosis and NAFLD-HCC.

## Figures and Tables

**Figure 1 cancers-13-00516-f001:**
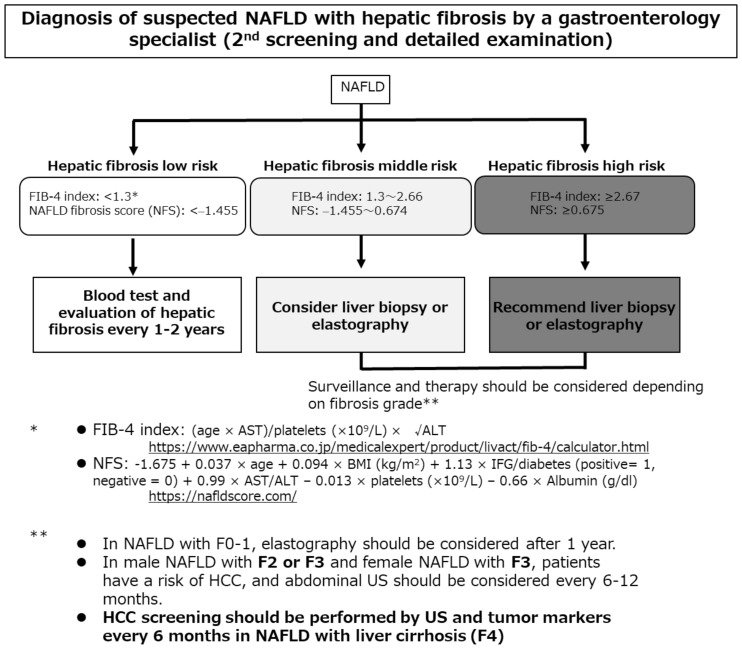
The algorithm for hepatocellular carcinoma screening and surveillance contained in the revised Japanese guidelines [6]. AST, aspartate aminotransferase; ALT, alanine aminotransferase; BMI, body mass index; FIB-4, Fibrosis-4; NFS, nonalcoholic fatty liver disease fibrosis score; IFG, impaired fasting glucose; NAFLD, nonalcoholic fatty liver disease; HCC, hepatocellular carcinoma; US, ultrasonography.

**Table 1 cancers-13-00516-t001:** Prevalence of NAFLD-HCC in patients with cirrhosis.

Author	Publication Year	Study Population/Follow-Up Period	Incidence of HCC Caused by NAFLD-Related Liver Cirrhosis	Findings
Bugianesi et al. [18]	2002	641 HCC patients with liver cirrhosis/1990–	44 patients had CC.	Obesity and diabetes were significantly more common in patients with HCC.
Marrero et al. [19]	2002	105 HCC patients/2000–	29 patients had CC; half associated with NAFLD (13 patients).	Fifty-three cases were detected by surveillance.
Malik et al. [20]	2009	98 NASH patients with cirrhosis underwent LT/1997–2008	17 (17%) developed HCC.	Survival after LT was 88% during 2.5 years of follow-up.
Yatsuji et al. [8]	2009	68 NASH patients with cirrhosis/1990-2006	11.3% developed HCC over 5 years.	
Hashimoto et al. [21]	2009	34 NASH-HCC patients/ 1990–2007	5-year cumulative incidence of HCC 7.6%.	Five-year survival 82.8%.
Ascha et al. [9]	2010	510 cirrhosis patients/2003–2007	195 patients with NASH cirrhosis of whom 25 (12.8%) developed HCC.	64/315 (20.3%) of HCV-cirrhotic patients developed HCC (*p* = 0. 03).
Tokushige et al. [22]	2011	14,530 HCC patients/2006–2009	NAFLD-HCC 2%. The prevalence of cirrhosis was 62% in NAFLD-HCC patients and 52% in those with “unknown” HCC.	ALC-HCC proportion 7.2%. The prevalence of cirrhosis was 78% in ALC-HCC patients.
Kodama et al. [23]	2013	72 patients with NASH cirrhosis patients/1990–2010	10/72 NASH-cirrhosis patients developed HCC. 10.5% over 5 years	6/85 ALD-cirrhosis patients developed HCC. 12.3% incidence of HCC over 5 years; ALD-cirrhosis patients.
Wong et al. [24]	2014	61,868 LT patients (10,061 HCC)/2002–2012	NASH-related HCC; 4-fold increase	
Tateishi et al. [25]	2015	5326 HCC patients with non-viral etiologies/1991–2010	NAFLD-HCC 596 patients (11.2%), of whom 368 (63.4%) had cirrhosis.	Child-Pugh classes A; 439 (76.5%), B; 120 (20.9%), C; 15 (2.6%).
Mittal et al. [26]	2015	1500 HCC patients/2005–2010	8% NAFLD 8% patients, of whom only 58.3% had cirrhosis	Patients with NAFLD-related HCC did not undergo HCC surveillance in the 3 years prior to HCC diagnosis.
Younossi et al. [27]	2015	4979 HCC patients/2004–2009	701 NAFLD-HCC patients (14.1%)	NAFLD-HCC exhibited a 9% annual increase. NAFLD increased 1-yr mortality: OR 1.21, 95% CI 1.01–1.45.
Kanwal et al. [28]	2018	296,707 NAFLD patients/ 2004–2008	490 developed HCC (0.21/1000 person- yrs). Annual incidence of HCC in cirrhotic patients 10.6 (range 1.6–23.7)/1000 person-yrs.	
Vilar-Gomez et al. [29]	2018	458 biopsy-confirmed NAFLD patients/1995–2013	<F3, 17%; 95% CI, 8–31% vs. F4, 2.3%, 95% CI, 1–12%	Annual incidence of HCC: Child-Pugh A5; 1.8, A6; 4.7.

ALD, alcoholic liver disease; CC, cryptogenic cirrhosis; CI, confidence interval; NAFLD, nonalcoholic fatty liver disease; NASH, nonalcoholic steatohepatitis; HCC, hepatocellular carcinoma; LT, liver transplantation; OR, odds ratio; yr, year.

**Table 2 cancers-13-00516-t002:** Prevalence of NAFLD-HCC in non-cirrhotic patients.

Author	Publication year	Study Population/Follow-UP Period	Incidence of HCC in Non-Cirrhotic Patients	Characteristics of HCC in Non-Cirrhotic Patients
Guzman et al. [30]	2008	50 HCC patients underwent explant treatment or liver resection/2004–2007	3 cases/5 NAFLD+5 CC cases lacked cirrhosis.	
Paradis et al. [31]	2009	128 surgically resected HCC patients/1995–2007	31 cases with MS were non-cirrhotic.	Some HCCs with MS arose via malignant transformation of a pre-existing liver cell adenoma.
Kawada et al. [13]	2009	1168 surgically resected HCC patients/1990–2006	8 cases (1%) were NASH, 6 were non-cirrhotic.	
Yasui et al. [32]	2011	87 NASH-HCC patients/1993–2010	F1-3; 43 cases	Liver cirrhosis was less common in males.
Ertle et al. [33]	2011	162 NAFLD/NASH-HCC patients/2007–2008	Non-cirrhotic 41.7%,	
Dyson et al. [34]	2014	632 HCC patients/2000–2010	Non-cirrhotic 31/136 (22.8%).	NAFLD-HCC was associated with a lower prevalence of cirrhosis (77.2%).
Perumpail et al. [35]	2015	44 HCC patients/2010–2012	Non-cirrhotic: 6 cases	
Mittal et al. [36]	2016	1500 HCC patients/2005–2010	Non-cirrhotic: 13%	
Mohamad et al. [37]	2016	83 NAFLD-HCC patients/2003–2012	Non-cirrhotic: 36 cases	HCC patients that were non-cirrhotic (compared to cirrhotic) were older (67.5 ± 12.3 vs. 62.7 ± 8.1 years); less likely to be obese (52 vs. 83%) or to have type 2 diabetes (38 vs. 8%); more likely to have single nodules (80.6 vs. 52.2%) of larger size (>5 cm) (77.8 vs. 10.6%); more likely to undergo hepatic resection (66.7% vs. 17%); and less likely to receive loco-regional therapy (22.3 vs. 61.7%) or DDLT (0 vs. 72.3%)
Gawrieh et al. [38]	2019	5144 HCC patients/2000–2014	11.7% were non-cirrhotic; of whom 26.3% had NAFLD	Older age, more commonly female, less frequently black. Larger tumors, less frequently fulfilled the Milan criteria, more frequently underwent resection, and experienced better overall survival than liver cirrhosis patients.
Bengtsson et al. [14]	2019	1562 HCC patients/2004–2017	NAFLD-HCC 225 patients 26.3% (14.4%), of whom 83 (37%) were non-cirrhotic	Older age, a lower prevalence of diabetes, and more frequent resection.Mortality was similar to that from liver cirrhosis.
Kodama et al. [39]	2019	104 NAFLD-HCC patients/2000–2016	F0-2; 35 cases F3-4; 69 cases	HCCs in non-cirrhotic patients were larger than in others and evidenced lower histological activity. The recurrence rate was significantly lower in NAFLD-HCC patients who were not cirrhotic (*p* < 0.01). Risk factors for recurrence were the male gender, lower serum albumin levels, and advanced fibrosis.
Tobari et al. [40]	2020	857 NAFLD patients/1991–2018	48 patients with non-cirrhotic and 71 with cirrhotic HCCs	Risk factors for HCC in non-cirrhotic patients were the male gender, light drinking, and a high FIB4 index.
Kanwal et al. [41]	2020	271,906 NAFLD patients/2004–2008	22,794 developed cirrhosis, and 253 HCC, of whom 64 were non-cirrhotic	The risk of HCC was 6.4-fold higher in patients with diabetes, obesity, dyslipidemia, and hypertension (HR: 6.42, 95% CI: 0.89–46.07).

CI, confidence interval; DDLT, deceased-donor liver transplantation; FIB-4, Fibrosis-4; NAFLD, nonalcoholic fatty liver disease; NASH, nonalcoholic steatohepatitis; HR, hazard ratio; HCC, hepatocellular carcinoma; MS, metabolic syndrome; vs., versus.

**Table 3 cancers-13-00516-t003:** Summary of NAFLD-HCC predictors.

Author	Publication Year	Study Population/Follow-Up Period	Diagnostic Method for NAFLD	HCC CumulativeIncidence	Risk Factors
Hashimoto et al. [21]	2009	382 NASH patients (34 NASH-HCC patients)/1990–2007	Biopsy	7.6%—5 yrs	Fibrosis, OR: 4.232, 95% CI: 1.847–9.698. Age, OR: 1.108, 95% CI: 1.028–1.195. AST, OR: 0.956, 95% CI: 0.919–0.995. Activity, OR: 0.154, 95% CI: 0.037–0.638.
Kawamura et al. [55]	2012	6508 NAFLD patients/12 yrs	Ultrasound; Fibrosis graded by biopsy in 104 cases	0.02%—4 yrs 0.2%—8 yrs 0.5%—12 yrs	F3/4, HR: 25.03, 95%CI: 9.02–69.52. DM, HR: 3.21, 95%CI: 1.09–9.50. AST level ≥40 IU/L, HR: 8.20, 95% CI: 2.56–26.26. Platelet count <15 × 10^4^/μL, HR: 7.19, 95% CI: 2.26–23.26. Age ≥60 years, HR: 4.27, 95% CI: 1.30–14.01.
Tokushige et al. [57]	2013	34 NAFLD-related cirrhosis patients/1990–2011	Biopsy	11.3%—5 yrs	Child-Pugh, HR: 3.09, 95% CI: 1.374–6.934. Serum GGT, HR: 1.01, 95% CI: 1.002–1.022. Age, HR: 1.12, 95% CI 1.014–1.226.
Seko et al. [58]	2017	312 NASH patients/1999–2014	Biopsy	1.9%—5 yrs 4.2%—6 yrs 8.3%—10 yrs	F3/4, HR: 24.4, 95% CI: 2.07–288.2. PNPLA3 genotype GG, HR 6.36, 95% CI 1.36–29.80.

BMI, body mass index; CI, confidence interval; DM, diabetes mellitus; FIB-4, Fibrosis-4; GGT, γ-glutamyltranspeptidase; NAFLD, nonalcoholic fatty liver disease; NASH, nonalcoholic steatohepatitis; HR, hazard ratio; HCC, hepatocellular carcinoma; OR, odds ratio; PNPLA3, patatin-like phospholipase domain-containing 3; yr, year.

**Table 4 cancers-13-00516-t004:** Summary of NAFLD-HCC outcomes after curative therapies.

Author	Publication Year	Study Population/Follow-Up Period	Treatment	Overall Survival	Disease-Free Survival	Other
Reddy et al. [67]	2012	52 NASH-HCC patients/ 2002–2010	DDLT, resection, RFA	1-yr: 90%, 3-yr: 72%, 5-yr: 65%	1-yr: 84%, 3-yr: 70%, 5-yr: 60%	
Wong et al. [24]	2014	61,868 end-stage liver disease patients (10,061 HCC patients)/2002–2012	DDLT	1-yr: 87.5% 3-yr: 79.8%, 5-yr: 65.5%	NA	
Piscaglia et al. [15]	2016	756 patients (145 NAFLD patients)/2010–2012	DDLT, resection, RFA	1-yr: 90–95%, 3-yr: 85–90%	NA	Survival was significantly shorter in NAFLD-HCC patients (25.5 months) (95% CI 21.9–29.1 months, *p* = 0.017).
Malik et al. [20]	2009	98 NASH cirrhosis patients (17 HCC patients)/1997–2008	DDLT	1-yr: 85–90%	NA	
Hernandez-Alejandro et al. [69]	2012	102 NASH patients (17 HCC patients)/2000–2011	DDLT	NA	1-yr: 95%, 3-yr: 95%, 5-yr: 85%	
Cauchy et al. [70]	2013	560 HCC patients (62 MS patients)/2000–2011	Resection	1-yr: 83%, 3-yr: 75%, 5-yr: 59%	1-yr: 83%, 3-yr: 70%, 5-yr: 66%	
Wakai et al. [71]	2011	225 HCC patients (17 NAFLD patients)/1990–2007	Resection			Patients with NAFLD exhibited better disease-free survival than did those infected with HBV or HCV.
Takuma et al. [72]	2011	36 cirrhosis-associated HCC patients/1992–2009	Resection, RFA, PEI, MCT	1-yr: 94%, 3-yr: 85%, 5-yr: 54%	1-yr: 89%, 3-yr: 68%, 5-yr: 54%	

DDLT, deceased-donor liver transplantation; HBV, hepatitis B virus; HCV, hepatitis C virus; MCT, microwave coagulation therapy; NA, not applicable; NAFLD, nonalcoholic fatty liver disease; PEI, percutaneous ethanol injection; RFA, radiofrequency ablation; yr, year.

## Data Availability

The datasets used and/or analyzed in this study are available from the corresponding author upon reasonable request.

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
