# Peer review of "The Current View of Nonalcoholic Fatty Liver Disease-Related Hepatocellular Carcinoma"

_cancers, 2021, doi:10.3390/cancers13030516_

Round 1

Reviewer 1 Report

This is an excellent review of the problem of HCC associated with NAFLD. The paper provides useful compilation of available clinical research on the prevalence of HCC in cirrhotic and non-cirrhotic NAFLD, the tables detailing these data are a great asset of the work. Although the review is studded with figures on HR, OR and RR relevant to the subject, it still reads well.

There are a few minor issues that may require attention. 

Line 77 - "Thus, HCC should be sought in non-cirrhotic cases in males" - this seems be too categorical as non-cirrhotic HCC is seen among females and in fact females may be dominantly affected in some cases (e.g., Phipps et al, Am J Gastroenterol 2020;115:1486). Thus, this summative statement should be rephrased.

Line 141 - In the chapter of Risk Factors, the following sentence seems to be a bit out of place: "In a systematic review, NAFLD and NASH cohorts lacking cirrhosis were at minimal risk of HCC development (cumulative HCC mortality 0–3% over study periods of up to 20 years)". It would be better to separate citations about cirrhotic and non-cirrhotic HCC, and put this line into the next paragraph.

Line 157 - The chapter "Treatment of NAFLD-HCC" discusses both prevention and treatment matters and could be perhaps better structured. Prevention by better diabetes control, use of metformin and other lifestyle/medication issues (alcohol use, weight management, statins) could be perhaps included in a separate chapter under "Prevention". 

Line 164 - The discussion of using sodium/glucose cotransporter 2 inhibitors does not seem to be relevant here, unless placed into an expanded prevention chapter as proposed above -  please consider removing it.

Line 208 - There seems to be some confusion here, coming from the statement: "At the first screening, serum markers of hepatic fibrosis (hyaluronic acid, type IV collagen 7S, and mac-2 binding protein) should be measured, fibrosis should be scored using the FIB-4 index or the non-alcoholic fatty liver disease fibrosis score (NFS), and an appointment to obtain a platelet count should be scheduled." First, hyaluronic acid etc. listed here are not really cheap tests and probably not needed if FIB-4 or NFS is considered in parallel, which are much cheaper and perform similarly well compared to more expensive and complex fibrosis scores. Second, it is not clear why an appointment for PLT count  would be needed, since PLT would be part of the FIB-4/NFS anyway and one assumes a CBC is being done along with the liver enzymes.

Line 221 - The following statement is a bit confusing: "Recently, the AGA proposed an HCC surveillance method for patients with NAFLD and liver cirrhosis, or severe fibrosis revealed by noninvasive testing [5]. At least two evaluation methods among [sic?] FIB-4 index calculation, enhanced liver fibrosis testing, FibroScan testing, and magnetic resonance imaging (MRI) are recommended." HCC screening/surveillance is widely recommended to occur by biannual echogram (or MRI) and tumor markers such as AFP may be added, but FIB-4 and other Fibrosis scores do not serve this purpose. Thus, it needs to be separated what screening test we recommend for fibrosis vs. HCC.

Line 234 - The chapter of Perspectives is somewhat of a hodgepodge and could be better structured. This should indeed include therapies when loco-regional treatment is not feasible any longer and/or discuss briefly personalized treatment, but could also discuss more the elephant in the room and what to do about it: how better to find at-risk NAFLD cases that would need HCC screening.

On this last note, Figure 1 could be optimized in some ways. This reviewer recognizes that this algorithm follows the current Japanese guidelines to be published by the senior author (reference could not be located yet). NAFLD at the top comes in many flavors, so perhaps the number and extent of metabolic risk factors co-occurring could weigh into the decision tree. Fibroscan follow-up in 1 year for patients with FIB-4 below 1.30 seem too generous (at least by US standards). Also, offering liver biopsy may seem too broad according to this algorithm - it is probably more important in the grey zone and to rule out concomitant liver disease. 

Author Response

We greatly appreciate the reviewers’ comments on our manuscript, entitled “The current view of nonalcoholic fatty liver disease–related hepatocellular carcinoma”. We have carefully revised the manuscript according to the suggestions made.

This is an excellent review of the problem of HCC associated with NAFLD. The paper provides useful compilation of available clinical research on the prevalence of HCC in cirrhotic and non-cirrhotic NAFLD, the tables detailing these data are a great asset of the work. Although the review is studded with figures on HR, OR and RR relevant to the subject, it still reads well.

There are a few minor issues that may require attention.

Line 77 - "Thus, HCC should be sought in non-cirrhotic cases in males" - this seems be too categorical as non-cirrhotic HCC is seen among females and in fact females may be dominantly affected in some cases (e.g., Phipps et al, Am J Gastroenterol 2020;115:1486). Thus, this summative statement should be rephrased.

We have changed to “In contrast, the frequency of HCC in non-cirrhosis was reported to be significantly higher in female than male (Phipps et al. 2020). Thus, HCC also should be noticed in non-cirrhotic cases”.

Line 141 - In the chapter of Risk Factors, the following sentence seems to be a bit out of place: "In a systematic review, NAFLD and NASH cohorts lacking cirrhosis were at minimal risk of HCC development (cumulative HCC mortality 0–3% over study periods of up to 20 years)". It would be better to separate citations about cirrhotic and non-cirrhotic HCC, and put this line into the next paragraph.

We have replaced this sentence to the next paragraph.

Line 157 - The chapter "Treatment of NAFLD-HCC" discusses both prevention and treatment matters and could be perhaps better structured. Prevention by better diabetes control, use of metformin and other lifestyle/medication issues (alcohol use, weight management, statins) could be perhaps included in a separate chapter under "Prevention".

We have added the chapter of prevention and described the effect of drug treatment.

Line 164 - The discussion of using sodium/glucose cotransporter 2 inhibitors does not seem to be relevant here, unless placed into an expanded prevention chapter as proposed above -  please consider removing it.

We have removed this sentence.

Line 208 - There seems to be some confusion here, coming from the statement: "At the first screening, serum markers of hepatic fibrosis (hyaluronic acid, type IV collagen 7S, and mac-2 binding protein) should be measured, fibrosis should be scored using the FIB-4 index or the non-alcoholic fatty liver disease fibrosis score (NFS), and an appointment to obtain a platelet count should be scheduled." First, hyaluronic acid etc. listed here are not really cheap tests and probably not needed if FIB-4 or NFS is considered in parallel, which are much cheaper and perform similarly well compared to more expensive and complex fibrosis scores. Second, it is not clear why an appointment for PLT count would be needed, since PLT would be part of the FIB-4/NFS anyway and one assumes a CBC is being done along with the liver enzymes.

This demonstrated in the Japanese guideline. “At the first screening, serum markers of hepatic fibrosis (hyaluronic acid, type IV collagen 7S, and mac-2 binding protein), or the FIB-4 index or the nonalcoholic fatty liver disease fibrosis score (NFS), or a platelet count should be evaluated. When liver fibrosis is indicated, consultation with a gastroenterology specialist should be considered for the second screening. Castera et al. [56] found that such simple, inexpensive, and widely available assays had high negative predictive values, effectively ruling out advanced fibrosis. Patients at low risk of advanced fibrosis (FIB-4 score < 1.3 or the NFS score < −1.455) do not require further assessment”.

Line 221 - The following statement is a bit confusing: "Recently, the AGA proposed an HCC surveillance method for patients with NAFLD and liver cirrhosis, or severe fibrosis revealed by noninvasive testing [5]. At least two evaluation methods among [sic?] FIB-4 index calculation, enhanced liver fibrosis testing, FibroScan testing, and magnetic resonance imaging (MRI) are recommended." HCC screening/surveillance is widely recommended to occur by biannual echogram (or MRI) and tumor markers such as AFP may be added, but FIB-4 and other Fibrosis scores do not serve this purpose. Thus, it needs to be separated what screening test we recommend for fibrosis vs. HCC.

We have changed the sentence to “When the quality of ultrasonography is suboptimal for screening of HCC, future Screening by either computed tomography (CT) or magnetic resonance imaging (MRI), every 6 months are recommended”.

Line 234 - The chapter of Perspectives is somewhat of a hodgepodge and could be better structured. This should indeed include therapies when loco-regional treatment is not feasible any longer and/or discuss briefly personalized treatment, but could also discuss more the elephant in the room and what to do about it: how better to find at-risk NAFLD cases that would need HCC screening.

We have added the possibility of personalized therapy.

On this last note, Figure 1 could be optimized in some ways. This reviewer recognizes that this algorithm follows the current Japanese guidelines to be published by the senior author (reference could not be located yet). NAFLD at the top comes in many flavors, so perhaps the number and extent of metabolic risk factors co-occurring could weigh into the decision tree. Fibroscan follow-up in 1 year for patients with FIB-4 below 1.30 seem too generous (at least by US standards). Also, offering liver biopsy may seem too broad according to this algorithm - it is probably more important in the grey zone and to rule out concomitant liver disease.

Although this figure is taken from the Japanese guidelines and cannot be changed, liver biopsy is mainly recommended in the cases of advanced fibrosis.

This manuscript was the review of “The current view of nonalcoholic fatty liver disease–related 2 hepatocellular carcinoma”

This paper review the NAFLD and HCC very clearly

we have some suggest to make the review completely

Reviewer 2 Report

Reviewers' Comments to Author:

This manuscript was the review of “
The current view of nonalcoholic fatty liver disease–related 2 hepatocellular carcinoma”
This paper review the NAFLD and HCC very clearly
we have some suggest to make the review completely

For Characteristics and pathogenesis of NAFLD-HCC section

The one of current important issue for pathogenesis was concerning about genetic problem. The author only mentioned the gene (PNPLA3) . As we knew, the HSD17B13 variant genotype was also associated the protection of NAFLD /NASH . We also found a lot of publications described the interplay PNPLA and  HSD17B13 variant to modulate the risk of HCC. We hope the author could make more detail review for this discussion

Another , we suggest author could make discussion for prevention or resolution of NAFLD/NASH. The relation between obesity and NAFLD was dominant. The most important issue is how to treat the NAFLD/NASH to prevent the progression of HCC. A lot of medical treatment and life style modulation were described. But also about bariatric surgery,  it is the most effective treatment for NAFLD and NASH for obesity patient. We hope the author could consider the another section to discuss the prevention and treatment of NAFLD/NASH.

Author Response

We greatly appreciate the reviewers’ comments on our manuscript, entitled “The current view of nonalcoholic fatty liver disease–related hepatocellular carcinoma”. We have carefully revised the manuscript according to the suggestions made.

For Characteristics and pathogenesis of NAFLD-HCC section

The one of current important issue for pathogenesis was concerning about genetic problem. The author only mentioned the gene (PNPLA3) . As we knew, the HSD17B13 variant genotype was also associated the protection of NAFLD /NASH . We also found a lot of publications described the interplay PNPLA and HSD17B13 variant to modulate the risk of HCC. We hope the author could make more detail review for this discussion

We have added the HSD17B13 in the genetic section. “Additionally, 17-beta hydroxysteroid dehydrogenase 13 (HSD17B13) was identified as having the protective role of NAFLD (Su et al. 2014) and protecting against HCC development (Chen et al. 2018). Lower HSD17B13 in non-cancerous tissues was associated with worse recurrence-free survival and overall survival in HCC patients. Although they mainly studied viral hepatitis-associated HCC, further study is required.”

Another , we suggest author could make discussion for prevention or resolution of NAFLD/NASH. The relation between obesity and NAFLD was dominant. The most important issue is how to treat the NAFLD/NASH to prevent the progression of HCC. A lot of medical treatment and life style modulation were described. But also about bariatric surgery,  it is the most effective treatment for NAFLD and NASH for obesity patient. We hope the author could consider the another section to discuss the prevention and treatment of NAFLD/NASH.

We have added the chapter of prevention and discussed about the bariatric surgery. Recent analysis demonstrated that bariatric surgery was reduced NASH-HCC [54]. Bariatric group showed lower incidence of new-onset HCC (0.05% vs. 0.34%, p=0.03) comparing to the propensity-matched control group. It might become an option for HCC prevention.